# Baroque Origins of the Greenery of Urban Interiors in Lower Silesia and the Border Areas of the Former Neumark and Lusatia

Bogna Ludwig 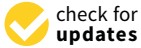

Department of Architecture Conservation and Restoration of Cultural Landscape, Faculty of Architecture, Wroclaw University of Science and Technology, 50-370 Wroclaw, Poland; bogna.ludwig@pwr.edu.pl

**Abstract:** The article is the first attempt to gather information on the beginnings of using green elements in urban compositions in Lower Silesia and border areas, in the former Neumark and Lusatia. It presents Baroque urban arrangements with the use of green ground floors, tree espaliers and avenues, from the earliest ones—occurring in the aftermath of the Thirty Years' War—and the solutions applied in private municipalities in the Habsburg, Wettin, and Hohenzollern states, which were recovering from war damage, to urban developments at the end of that period, in the areas already under Prussian rule and its strict regulations. A comparison with the achievements of European urban planning in this field allows us to trace the paths of inspiration, but also to uncover some innovative achievements.

**Keywords:** urban baroque greenery; 17th–18th century; allée; Lower Silesia; preservation and renewal of heritage



## 1. Introduction

Although there were no large or dynamically developing urban centers in Lower Silesia during the Baroque period, there were changes in the appearance and methods of design of architecture, which largely influenced the appearance of municipalities, changes which were based on a modern view of the relationship between public space and buildings. Greenery complexes were among the new components introduced to Baroque spatial arrangements of urban interiors. These urban planning activities have rarely been noticed by researchers, as a result of which no comprehensive scientific study has yet been devoted to them yet.

Greenery was only introduced into urban planning arrangements when the residential municipalities, still autonomous of the Habsburg State, were transformed in the Duchies ruled by the Houses of Piast and Württemberg as well as in Albrecht von Wallenstein's estates. The investments of Louise of Anhalt-Dessau (1631–1680), Christian Ulrich Württemberg (1652–1704), and his wife Anna Elisabeth of Anhalt-Bernburg (1647–1680) were of particular importance. In the following period, new small private urban centers were established in which the owners of the border lands received religious refugees from Silesia, Bohemia, and Moravia; in such municipalities, tree avenues made up the center or one of the most important elements of the composition. The initiatives of Count Balthasar Erdmann von Promnitz (1656–1703), Konrad von Troschke (1671–1728), and then the extensive actions of Nikolaus Ludwig von Zinzendorf (1700–1760), resulted from economic reasons, but they also clearly had an ideological basis. The last villages with Baroque layout and greenery compositions were created in the Prussian state after the borders had been shifted. At the same time, according to the new principles of fortification design, tree espaliers became a component of fortification systems and also served to protect the roads, which contributed to their further dissemination as a composition element in urban planning.

The composition of greenery as a new urban solution was a trend that municipality owners took over from other regions of Europe. For this purpose, they hired suitable contractors. This phenomenon was initially clearly related to the descent and connections of

ducal families in Lower Silesia. In the second period, the planning solutions were modeled primarily on those that were being developed in the immediate vicinity. Inspirations were also drawn during trips to remote European centers.

## 2. Materials and Methods. State of the Art and Sources

The study was based on a typical research method used in the humanities. Based on a review of the scientific literature on baroque urban greenery in Europe, the research background was characterized. The current state of knowledge was presented, and in some cases an attempt was made to supplement it using the analysis of sources. A review of scientific literature concerning the chosen subject of research has been made. The identification of urban greenery systems in Lower Silesia and neighboring areas created in the Baroque period was based on a meticulous analysis of available archival documents—cartographic, iconographic, and written documents—as well as chronicles and topographic descriptions. The few preserved greenery complexes in the discussed area arranged in the Baroque period were examined. The historical background of their creation is presented; the founders and—where possible—designers are indicated, with an attempt to trace the origin of inspiration. Their further development is briefly characterized. The description and formal analysis of the presented urban greenery structures was based on source information and the remains present in the urban layout. In this way, the results of the research were formulated in a chronological and typological sequence of urban greenery in Lower Silesia and the border areas of the former New March and Lusatia in the Baroque period. This allowed comparisons to be made with European examples, indicating the typicality of solutions, the use of selected patterns, and—in some cases—innovation.

Research on the history of greenery design in Europe [1] shows the genesis and dissemination of solutions from modern times for introducing greenery composition into urban interiors. In this respect, it is important to indicate why and how the ideas of designing greenery systems, initially only tree espaliers and avenues, used in gardens and open landscapes, were transferred to inner municipality structures. The juxtaposition of the emerging pan-European concepts in this respect allows us to refer to them realizations from the presented region.

Baroque urban planning in Lower Silesia and the border areas is a rarely studied issue. Therefore, it is difficult to expect research on selected, quite detailed problems, such as the occurrence and the way of shaping greenery in public interiors. Studies of the historical spatial transformations of individual municipalities—from the oldest publications from the turn of the 20th century [2,3] to contemporary ones [4–6]—reveal the characteristics and sometimes also the landscape role of Baroque transformations. These publications also include information on the introduction of greenery systems to urban design. However, the very phenomenon of the greenery of urban interiors occurring for the first time in the Baroque period has not been properly noticed. This issue is partially discussed in an article devoted to the history of gardens from that period, which were also used as types of parks established at the settlements of the Moravian brothers in Lusatia, Hessen and Lower Silesia [7]; yet, the aforementioned text does not characterize the phenomenon of urban interior design with greenery systems. Similarly, the book by the same authors mainly mentions the greenery of gardens and parks, not urban greenery [8]. Historical research, in which the landscape role of the avenue is discussed, concerns the later assumptions and usually palace-park complexes [9]. In recent years, there have been fairly extensive studies on the natural qualities of avenues, which is a completely separate theme.

The analysis of local sources makes it possible to trace when and in what forms the first elements of greenery that shape urban compositions in Lower Silesia and the border areas of the Neumark and Lusatia emerged. The most precise information on this subject is provided by iconographic representations, made shortly after their creation, such as drawings by F.B. Wernher [10–16]. Cartographic records coming only from the 19th century are a valuable supplement of date [17–22]. Notes from Baroque chronicles written by F. Lucae [23] and the already mentioned F.B. Wernher [11–13], as well as from

18th and 19th century historical and topographical descriptions of Prussian Silesia by F.A. Zimermann [24] and J.G. Meissner [25], as well as descriptions of the March of Brandenburg by F.W.A. Bratring [26] and Saxony by F.A. Schumann and A. Schiffner [27] are important sources of knowledge.

From these juxtapositions, one can read out what patterns and how quickly they were adopted in the discussed area, answer the question of whether own solutions were developed and to what extent this new form of space arrangement using greenery composition was widespread.

## 3. Results

### 3.1. Avenues and Tree Espaliers—The Beginnings of Urban Green Space in European Municipalities

Since the end of the 17th century, trees have been consciously introduced as compositional elements, first in the form of tree espaliers and then allées. In ducal municipalities, the tree espaliers began to mark important spatial connections between the ruler's seat and the temple. Then, they became widespread in the municipalities possessed by the landed aristocracy as well. The tree espaliers began to be used as a composition element during the Renaissance. Allées appeared in garden arrangements. Initially, the straight paths were covered with pergolas overgrown with vines to provide shade (A1 in Appendix A) [28]. Such solutions were already applied in the Middle Ages. The name espalier (Italian: *spalliera*) is derived from that period. Alberti in the Renaissance villa Quaracchi Giovani Ruccelai (1459) divided the grove—*boschetto*—with straight paths [29,30]. This method of design quickly spread in the sixteenth century and became the principle of shaping the gardens of the Italian country villas (Villa Medcici in Castello, in Pratolino, the Boboli Gardens by the Pitti Palace, the villa in Poggio a Caiano, Poggio Reale near Naples) and the gardens by the residences all over Western Europe, especially the hunting parks (Blois, Gaillon, Hampton Court). It was also then that the term allée was introduced to name such paths (A2 in Appendix A) [31,32]. Wooden avenues were shaped outside the garden systems as paths leading to them (Chenoncaeux, Fointenbleau, Blois, Morienmont, Aranjues) [1] (pp. 18–19). Palladio in his treatise recommended providing country roads with tree espaliers giving shade and constituting greenery as a respite for eyes (A3 in Appendix A) [33] (pp. 262, 266). He also gave examples of the few such roads to Villa Cicogna and Villa Quinto in its area.

At the same time, rows of trees planted over moats and canals began to become a decorative element of the municipality. At the end of the 16th century, Dutch municipalities such as Bruges, Gouda, Harlem, The Hague, Amersfoort, and Kampen had wooded canal piers [34–37]. In The Hague, at the inner lake, Hofvijver, which was regulated by a rectangular basin, tree rows and greenery systems appeared as early as in the 14th century [38,39]. The wooded area of Lange Voorhout was incorporated into the municipality at the beginning of the 15th century. In 1536, during his visit, Emperor Charles V ordered the planting of an avenue of linden trees, which would connect the adjacent gardens in this area [40–42]. Trees were also planted in the 1670s around the fortifications in Antwerp, and at the end of the 16th century also in Italian municipalities of Florence, Siena, Piacenza, Lucca, and Padua. The enlargement of municipalities also boosted the appearance of groups of trees. In Amsterdam, tree-lined canals were introduced in place of old fortifications after they had been shifted to a new line in 1585 and once again in 1610 [43]. In Leiden, such changes occurred slightly later, and in Utrecht in 1660 [1] (pp. 44–45). In the 17th century, the custom became widespread and tree-lined canals were created in many Dutch municipalities.

In accordance with Palladio's recommendations, roadsides were lined with trees from the end of the 16th century. The suburban roads of Rome have mostly turned into avenues over several decades [44,45]. In France, already in 1553 and 1575, kings Henry II and Henry III ordered the royal roads to be tree-lined. However, it was only J.B. Colbert's regulations of the 1650s, which applied the 1601 Edict of Sully, that decided that this principle was fully implemented [1] (p. 19) (which, for example, was clearly visible around Reims [46]). Outside Italy, roads leading to residences were also designed in the form of avenues. The earliest example of which is the tract to Hellbrunn from Salzburg, created

in 1615–1619 [47,48]. In the 17th and 18th century, avenues leading to palaces became the norm (e.g., from the more famous Fürstenallee in North Westphalia to the hunting lodge Oesterholz, 1725, Poppelsdorferallee near Bonn to the palace of the same name, around 1730, Wilhelmshöher Allee in Kassel, from 1767, and Laxenburger Allee and Schönbrunner Allee near Vienna, 1741).

Social changes in big municipalities—the settlement of courtiers and nobles, as well as the growth of a rich patrician group—changed the habits and the manner of having entertainment. Horsemanship, outdoor games, such as tennis, golf, various ball games, pall-mall (paille-maille), and shooting galleries, required green areas. Suburban recreation areas began to emerge; in the Renaissance, suburban gardens (e.g., in the neighborhoods of Florence, Rome, Naples, Paris, London, Bruges, and Antwerp), in the Baroque were replaced by promenades and suburban avenues and boulevards. Special areas for pall-mall appeared from the end of the 16th century in Paris, The Hague, Utrecht, and London. Then, they were often used as walking promenades and as such were copied in other municipalities in the 17th and 18th centuries. Riding avenues, the route over the Arn near Florence and the Course la Reine on the Seine, modeled on it in Paris (1616), inspired further routes in the vicinity of Marseille and Aix, but also outside France near Madrid or London [1] (pp. 32–37).

The first urban interiors designed using tree espaliers of trees were also realized. In Aix en Provence, the square of the new district of Villeneuveve was lined with trees in 1580. In Willemstadt and Klundert, the new fortress municipalities founded in the 1580s in North Brabant by Wilhelm of Orange, trees surrounded some squares and streets [1] (pp. 22–26) [49,50]. Created in 1630, Covent Garden Piazza in expanding London gave rise to squares as centers of aristocratic districts—Lincolns Inn, Bloomsbury Square, St. James Sq. in 1650–1660—which were initially only lined with by tree espaliers and lawns. Others appeared after the great municipality fire (1666) and became a model for other English municipalities; e.g., Warwick or Bristol. In the second half of the 17th century, walking avenues were also created in English municipalities; leading to the cathedral in London, in Bristol, Bath, Wells, and many provincial centers [1] (pp. 38–51). In Rome, from the beginning of the seventeenth century, elm tree avenues were introduced at the passageways: connecting the basilicas of Santa Croce and San Giovanni and further on Santa Maria Maggiore, the Arch of Constantine with the church of San Gregorio al Celio, as well as at Campo Vaccine on the site of the cattle market at the Roman Forum recreating the Via Sacra between the arches of Titus and Sever (A4 in Appendix A) [51]. Within the precincts of Paris there were established the royal gardens of Thulleries (created from 1564) and Luxembourg (completed in 1612), in the vicinity of Place Royale (later Place des Vosges, also formed until 1612) along with the gardens of Palais Royale (in 1633), and then the boulevards created on the site of the fortifications (Les Grands Boulevards, 1668–1705). However, public greenery appeared inside the municipality sporadically, only in the form of the greenery of Vosges Square in 1680. Further expansion of the metropolis connected it with suburban green areas such as Les Invalides Square (1670–1676), Jardin des Plantes (1626, as a public garden since 1640), or the Champ de Mars (1751) [52]. The tree arrangements in the urban composition did not gain a significant share in the Baroque in France. Even in the second half of the 18th century, when e.g., the extension of Rouen was planned, the project made by M. le Crapentier (1758), which included avenues and squares surrounded by tree espaliers, was mostly not realized; some of the works started only at the very end of the 18th century. Similarly, two proposals for the extension of Lyon (1768 and 1769) containing analogue concepts were rejected [1] (p. 66).

It was different in the expanded and established municipalities in Prussia and German countries, where Dutch examples of the use of tree espaliers but also French garden solutions were applicated in shaping urban layouts. The Hague Voorhout became an inspiration for an avenue, initially simply called Erste Straße, which was created on the initiative of the Great Elector Frederick William I in Berlin. It was planted as early as 1647 according to the Dutch model, with lime trees (hence from 1734 called Unter den Linden) [53]. The avenues

formed the urban backbone of the New Municipality in Dresden; first, the main axis of the Hauptstrasse (1687) was formed and then extended along the Koenigsstrasse (1722–1732). In Frankfurt, in 1712 the main square of the New Municipality—Rossmarkt—was, on the order of the municipality council, provided with greenery around the fountain and a chestnut allée extended therefrom along the frontage of new houses [54]. The greenery compositions had the greatest share in the planning of the new residential municipality of Charles III Wilhelm von Baden-Durlach—Karlsruhe—in 1715, often regarded as a manifestation of Classicist urban planning. This architectural concept was based on a pattern of radial tracts spreading from the palace (or in fact the tower next to it), formed as the streets continued by the garden avenues. Similarly, one of the two axes shaping the composition of St. Petersburg (founded in 1703)—Nevsky Prospect—featured tree espaliers [55]. The avenues also became the basis of the layout of municipalities and districts for Huguenot refugees, as was the case Potsdam. Potsdam became a residential municipality from around 1675, intensively populated by Huguenots emigrating from France after the Fontainebleau and Potsdam Edict from 1685. It developed along the allée stretching from the elector's hunting lodge into an open area. The Huguenots' refugee districts built near the capital municipalities of Erlangen, Kassel, Magdeburg, and Plön were similarly planned. For example, the new municipality in Erlangen, designed by Johann Moritz Richter, builder of margrave Christian Ernest Hohenzollern-Bayreuth (1686, 1706), together with the construction of the palace from 1700 onwards, took on the layout of a municipality-residence with a palace-garden composition axis. The avenues also surrounded and connected new districts of a different character, such as the Herrenhauser—residential district of Hanover. Chestnut promenades connected the expanding Bayreuth (1725) and Ansbach (1737) [56,57]. Tree espaliers also began to be used to emphasize the representativeness of the site, as in Manheim around the monument on the Pardeplatz designed in 1712 (made in 1743) to commemorate the victory in the war of succession in the Palatinate (1688–1697).

The problem of trees was treated somewhat differently by Frederick William I (1688–1740) and then Frederick II the Great (1712–1786), also taking into account their economic significance. It was recommended to plant fruit trees, mulberries for silkworm rearing and willows for wicker production. In time, state orders were issued, initially covering the protection of planting, and subsequently orders for the introduction of avenues of trees (A5 in Appendix A) [58,59]. In 1742, the planting of fruit trees and mulberries was ordered, which also covered Silesia, conquered at that time [60–63].

### 3.2. Greenery of Urban Ducal Residences in Lower Silesia

As in Western Europe, the initial introduction of greenery into municipalities in Lower Silesia was associated with the transformation of the ducal urban seats, while on their outskirts with the formation of links between municipalities and neighboring and suburban palaces. In modern times, greenery compositions were gaining in importance in residential layouts, initially only for recreation and decoration, then also for composition.

The visual impact of residences on society began to become increasingly important in modern times. For this purpose, different means were made use of than those in the Middle Ages, where monumentality and inaccessibility were the dominant features of a ruler's residence or a feudal lord. In modern concepts, the criterion of representativeness expressed through classical order architecture but also through the surrounding greenery came to the fore. In addition to the architectural changes in line with the current trends, it was the garden facilities that gave prestige to the ruler's seat, as evidenced by the castle gardens in Brieg (currently Brzeg) and Glogau (Głogów). The Brieg castle was transformed and thoroughly rebuilt during the times of Duke Frederick II (1480–1547) and his son George II (1523–1586), and thus changed from a Gothic defense building into a magnificent Renaissance residence [23] (pp. 1363–1364) [64] (pp. 2–4) [65] (pp. 25, 28–40) [66]. From the foundation of George II and his wife in 1554–1560, a three-story building of the entrance gate, the most important investment shaping the appearance of the castle from the municipality side and presenting a rich iconographic program, was

erected between the south wing and St. Hedwig's collegiate chapel [67]. At the same time, the collegiate church was transformed into a mausoleum of dukes. Italian architects Jakob and Franziskus Pahr were employed in the reconstruction of the castle. The continuators were Bernard and Peter Niuron, who also fortified the residence at the end of the century. The surroundings were also changed to shape the castle's representative gardens. To the north of the castle, an orchard was planted along the walls, which incorporated the former Dominican monastery grounds, with a hill (Sperling Berg) [68] (pp. 77–78) (A6 in Appendix A). On the east side, a garden (*Lustgarten*) was arranged in front of the windows, covered with a wall on the municipality side Figure 1a.

Similar transformations were made in the Glogau castle. At the turn of the 15th and 16th centuries, Glogau was inhabited by Prince John Albert, and then by Sigismund Jagiellon (both after Kings of Poland and Grand Dukes of Lithuania), who began rebuilding the castle, which had been ruined during the siege by Matthias Corvinus' army. Gardens were then arranged next to the castle, while behind the bailey and the second line of moats and walls, an entrance square was created [69].

In the Baroque era, residences and gardens began to be composed in a new way. Activities were undertaken both in the ducal residences and the former manors of the landed gentry which were rebuilt into palaces. In exceptional cases, they were matched with the municipality layout, as was the case in Juliusburg (Dobroszyce) or Festenberg (Twardogóra). After the Thirty Years' War, not only different Baroque architectural and ornamental forms were used, but also the approach to space arrangement changed. Compositions with a dominant axis were introduced. In representative forms, the foreground of the residence was designed. Extensive garden arrangements with pavilions, orangeries and loggias were created.

The first gardens of this type were established by the order of Louise of Anhalt in the 1670s, in the Duchess's seats in Brieg, Ohlau (Oława), and Wohlau (Wołów), which were being rebuilt at that time. As Baroque chronicler Lucae noted, these gardens were to provide a beautiful vista [23] (p. 1374). In all castles, the Duchess introduced terraces and vista points to the greenery of the gardens. In Brieg and Wohlau, it was probably mainly about views from the palace windows to the garden, but in Ohlau the front gardens also provided an attractive view of the new residence from the municipality side. The green areas thus became the backdrop for the ducal residence.

In Brieg, the transformation of the residence began as early as during the times of Duke George III (1611–1664). In 1654, George took over the Brieg Duchy [70]. Already in 1656, he started the reconstruction of the Renaissance castle in Brieg. One of the first investments was to clean up the foreground on the municipality side. An openwork fence was erected to separate the courtyard and the castle garden from the municipality square (Topffmarkt), to which a gate with wickets, situated in front of the gate tower, led, which was shaped in the Palladian form, being a kind of triumphal arch [23] (p. 1363) [11] (pp. 297–301). On the western side, behind the fence, partly in front of St. Hedwig's Castle Chapel, in 1658, there were farm buildings and a riding school, forge, and stable, forming a screen around the castle. The front garden (*Lustgarten*) was redesigned. In the following year, the castle façades and all towers were renovated after the Lions' Tower, rebuilt in 1649, was destroyed by thunder. Already during the reign of the next Duke Christian, garden buildings were erected in the orchard; a loggia with an orangery, a birdhouse and a shooting range (A7 in Appendix A) [8] (pp. 156–157). The castle building was rebuilt to open from the east side to the garden with arcades [23] (pp. 1364, 1367, 1378, 1380) [64] (pp. 23–25).

Further Baroque changes were ordered in 1673 by Duchess Louise of Anhalt, widow of Duke Christian, George's brother. The bay window on the eastern façade of the castle was then extended; it was crowned with a terrace and enlarged while its windows were made uniform, and a new arcade loggia was constructed on the municipality and garden side. In this way, the Brieg Castle received not only a representative entrance square on the municipality side and a decorative front garden but also a vantage point Figure 1b.

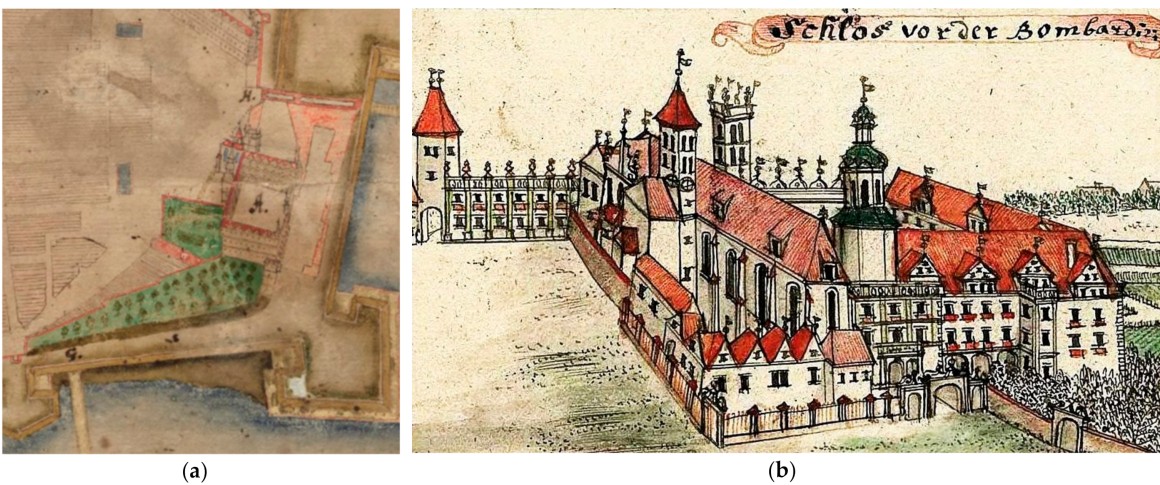

**Figure 1. (a)** The ducal residence in Brieg. A fragment of the Brieg plan, probably from the period between 1618 and 1633 [71]; **(b)** The ducal castle in Brieg around 1750 in a drawing by F.B. Wernher [11] (p. 297).

As a regent, the Duchess held power in the Liegnitz-Brieg (Legnica-Brzeg) Duchy in 1672–1675 because her son George IV William (1660–1675) was underage. After the death of George William, who reigned for only a few months, the duchy came under direct rule of the Habsburgs and the Duchess settled in Ohlau. The Brieg castle used by Austrian officials and visiting dignitaries was partially neglected [68] (p. 82). The Jesuits soon located their seat next to it and occupied most of the former Dominican monastery grounds, which was to become the castle garden. In 1741, the Brieg residence was shelled and burnt down by the Prussian army. The façade of the entrance building and part of the ground floor of the eastern wing survived [64] (pp. 16–17). It was rebuilt into an inn and warehouses (1743–1755), while its architectural form was no longer taken care of.

In the modestly presented urban development of Ohlau, the castle's representative foreground was also taken care of. An original layout was created here, consistent with the municipality's location plan, which included a Baroque urban interior combining two areas of urban and residential function. In 1654, after the death of Duke Georg Rudolf, the lands of the Duchy of Liegnitz were divided among the sons of his brother Johannes Christian, Duke of Brieg [70]. The youngest of them, Christian (1618–1672), received Wohlau and Ohlau. He had Carlo Rossi from Como start the reconstruction of the dilapidated Ohlau residence in 1659 [71]. The works under the direction of the Italian architect lasted until 1680, well into the time of Louise of Anhalt [23] (pp. 1405–1406), whom the emperor granted the Ohlau and Wohlau lands for life. At the same time, the castle changed the defensive system of the roundel into an earth bastion. Rossi modernized the residence itself [72] (pp. 43–53) [73] (p. 383). In the years 1659–1673, he erected or converted the former building, which was added to the residential section on the east side (now a church stands there), into a two-story pavilion in early Baroque forms topped with a terrace with figures [11] (p. 398–400). This part of the castle was henceforth called *Christianbau*. From 1655, a chapel on the northern side of the courtyard began also to be rebuilt and a garden behind the palace. In the further 1673–1680, an extension of the castle, a four-story part called Duchess Louise's palace (*Louisenbau*), was created, which opened from the courtyard with an arcade loggia, and was decorated with Baroque details on the front complete with a richly decorated portal depicting the Duchess's coat of arms [23] (p. 1406). In front of the residence, the foreground was cleaned; the buildings were demolished, the bridge was moved, leaving the moat, the castle fortifications on the municipality side were removed, and a garden was established in their place. A decorative fence with stone posts along the palace façade was introduced, separating the green belt from the moat on both sides of the bridge, Figure 2 [11] (pp. 267, 396–397, 400) [74] (A8 in Appendix A) [75–78]. A drawbridge, preceded by stone pillars-pylons, leading to the main entrance portal to the inner courtyard of the mansion was built in the middle of it (A9 in Appendix A). The square's compositional

axis was arranged, directed at the portal on the palace's façade, which was deliberately moved from the building's axis of symmetry to the east and emphasized by the use of double windows and the emerging castle tower behind it. This axis was marked out on the axis of the location municipality, i.e., in the middle of the market quarter facing the Castle Square, Figure 3. Its continuation in the open landscape was the main avenue of gardens behind the moat and bastion fortifications, highlighted by a pavilion in the middle [11] (p. 397) [23] (p. 1402). In this way, a very coherent and homogeneous composition of this urban layout was created, which obliterated the various origins of the elements of the location municipality and the medieval castle being transformed into a palace.

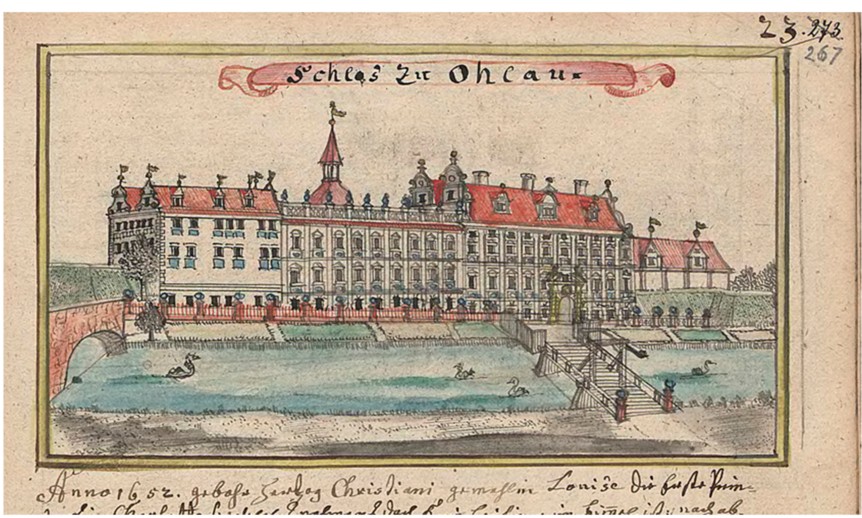

**Figure 2.** The southern elevation of Ohlau Castle on a drawing by F.B. Wernher from around 1730 [11] (p. 267).

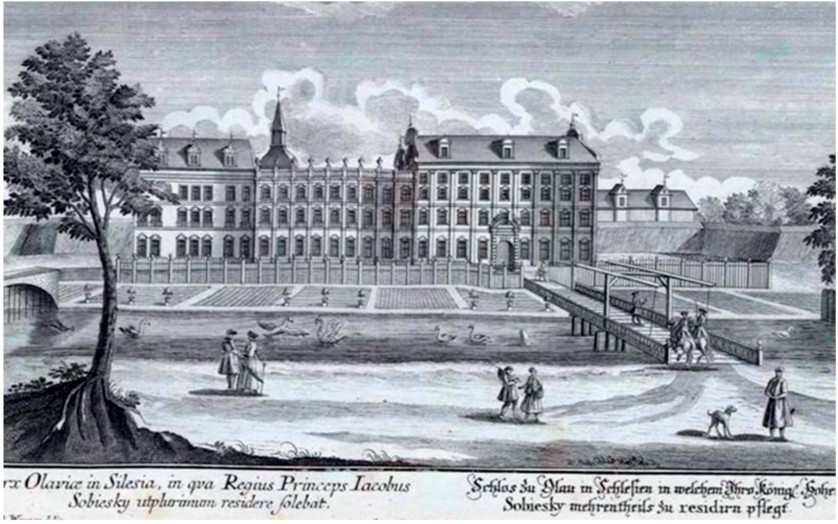

**Figure 3.** A view of castle buildings and gardens in a drawing from the times of James Louis Sobieski [78].

In 1691, Prince James Louis Sobieski settled in the castle in Ohlau to live there for over forty years. After his 1734 move to Żółkiew, the building began to deteriorate. It was taken over by the imperial court after Sobieski's death (1737) and became the residence of imperial administration officials. After Silesia had been incorporated into Prussia, a military lazaret was installed in the castle in 1744 and a field bakery in 1761. The moats were eliminated whereas the gardens were expanded. Rows of mulberry trees were planted.

From 1764, next to the royal office, there was a silk spinning mill in the castle. In the years 1833–1834, a part of the castle (*Chrystianenbau*) was demolished and a Catholic church designed by Karl Friedrich Schinkel was erected in its place; after a 1938 fire, a modernist tower was added to it. In Louise's building, a school was built, next to which a small barracks guardhouse was erected and at the beginning of the 20th century a water tower, thus degrading the completely representative character of the castle complex [79]. The castle square in the second half of the 19th century was turned into a green square with a monument to the Victory in the War of 1870 (Siegesdenkmal).

In Wohlau, ravaged by the Thirty Years' War, the castle was repaired and decorated in a new Baroque style. A representative bridge leading to the mansion over narrow moats was also built [23] (pp. 1162–1163) [80]. It is very probable that already then, when demolishing the serviced buildings of the castle and the north-western corner of the market quarter, an entrance square was marked out from the municipality side. Along with these regulations, a modest green ground floor in front of the castle façade was perhaps introduced. They certainly existed already after the reconstruction ordered by the managers of the Wohlau camera, located in the residence, when after the death of the Duchess (1680) the land came under the direct rule of the imperial court [11] (pp. 562, 569).

In Bernstadt (Bierutów), like in Ohlau, during the times of Christian Ulrich of Württemberg (1652–1704), front gardens were established on the foreground of the duke's seat after the land had been cleared. Bernstadt from one of the subordinate urban centers in the Duchy of Oels (Oleśnica) changed in modern times into the seat of the side lines of the ruling family. The owners of the residence and the municipality tried to give it attractive external features, raising the importance of their family. As soon as Henry II (1507–1548), the third of the House of Poděbrady, took the reins of power, he chose the Bernstadt castle as his residence. In the years 1534–1540, he carried out thorough reconstruction and gave the castle Renaissance forms. In 1603, a fire wreaked havoc with the castle and the whole municipality. The eastern wing and tower burnt down, and reconstruction lasted for the next few decades [81] (pp. 516–519) [82] (pp. 22–30) [73] (pp. 530–533). In spite of that, from 1618, Duke Henry Wenceslaus of Poděbrady lived there (1592–1639, Duke of Bernstadt in the years 1617–1639, and in the years 1629–1639 the General Governor of Silesia). The Bernstadt residence was a simple two-winged building (with southern and eastern buildings) with a preserved Gothic tower at the end of the eastern wing. Three-story arcade porches were constructed at the wings, giving it a Renaissance character, and in 1622, the roof on the tower was replaced by an early Baroque glorious helmet. At the same time, the municipality, which gained the status of a ducal capital, was changing intensively and was expanding despite fires (1603, 1659). After the Württembergs took over the Duchy of Oels in 1647, Bernstadt was, after the fire of 1659, an insignificant ducal seat [80,81] (p. 467). However, in 1673, when the duchy was divided, it became the residence of Silvius Nimrod's second son, Christian Ulrich. The eldest son, Silvius Frederick (1651–1697), received the Oels part, while the youngest, Julius, the Juliusburg part, which was ruled by Elizabeth (1625–1686), as he was a minor until his death [83].

Christian Ulrich resided in Bernstadt until the death of the oldest of his brothers, Silvius Frederick, in 1697, when he inherited Oels. During this period, he took steps to modernize the castle. He expanded it by constructing one story more. He also tried to raise its representativeness by changing the form of the avant-corps of the south wing, introducing there a Baroque portal decorated with the family coat of arms, while at the same time renovating the tower and crowning it with a helmet (1679–1680) [81] (pp. 469, 521–522). On the municipality's side, in the wall closing the castle courtyard, a decorative gate in the form of a palladiana was inserted around 1680, and in front of it a bridge over the moat preceded by a small entrance square (A10 in Appendix A). A larger viewing foreground was probably to be ensured by replacing the last quarter adjacent to the castle with a riding arena, Figure 4 [13] (pp. 688–695). Gardens were to be an additional decoration and attraction; a small Lustgarten with an orangery on the northern side of the castle, a castle garden on the southern side, and extensive gardens set up behind the municipality

walls in the immediate vicinity of the residence [81] (p. 521). These were realized under the close supervision and active collaboration of the Duke and his gardener Georg Herbst, who described the achievements in a work published in Oels [84]. The main entrance of the castle gate was located on a long viewing axis opening from the market square, via Breslauerstrasse, a view of the castle tower. From another point, the coat of arms portal of the main wing of the castle is visible through a side passage of the gate [85]. A line of trees was introduced parallel to the wall covering the garden. (Judging by Wernher's drawing, these were linden trees.) The espalier ran along the road connecting the gate with the church, or in fact the side gate in the church wall, which the Württemberg family used to go to the Duke's lodge (founded by Duke Christian Ulrich in 1679, as confirmed by the coat of arms of Württemberg and his wife Anna Elizabeth of Anhalt-Bernburg).

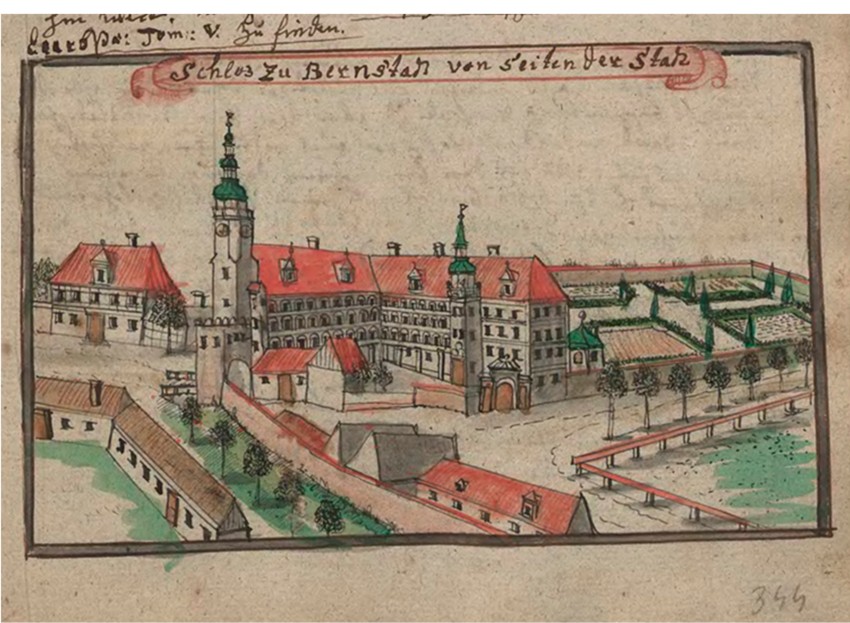

**Figure 4.** Bernstadt Castle illustrated by F.B. Wernher [13] (p. 691).

From the Baroque arrangement, only the portal of the entrance gate and the portal of the main castle wing have survived to this day; there, the coats of arms commemorate the founders. In the panorama and views, the presence of the castle is still signalled by the tower with its magnificent Baroque coat of arms. In the foreground of the residence on the municipality side there is still greenery, no longer a regular line. A similar but even more modest arrangement of the entrance to the castle from the municipality side was realized in Praisnitz (Prusice), where only a short line of trees preceded the elevation of the residence [11] (p. 651).

The gardens surroundings of the residences founded by the princes of Oels were the best witnesses of the importance of greenery in functioning of their courts. All three successive wives of Duke Christian Ulrich received new palace buildings with extensive gardens. Maria Sybillia received the palace in Sybillenort (Szczodre) and Sophia Wilhelmina, the palace in Wilhelminenort (Brzozowiec). In Oels, the gardens were established only for the fourth wife of Duke Christian Ulrich—Sophia of Mecklenburg at the beginning of the eighteenth century. In addition, his sister-in-law, Silvius Frederick's wife—Eleonore Charlotte of Württemberg-Montbéliard (1656–1743)—arranged a garden in Festenberg surrounded by rows of trees on the outskirts of the municipality [13] (pp. 740–741). The impossibility of introducing green complexes in the immediate vicinity of the residence was compensated for by creating separate suburban gardens. In Liegnitz, the gardens of Duchess Anna Sophia of Mecklenburg were created at the Sophienthal manor house (1657) and behind the Goldberger (Złotoryjska) Gate (1672) [23] (p. 1199). In Bernstadt, apart from

the gardens within the municipality walls to the north and south, a large garden complex was also set up outside the municipality walls in the immediate vicinity of the castle.

The compositional significance of trees in the urban layout was planned to be used in Silesia probably already before. Perhaps already in the first half of the 17th century, it was intended to link the layout of the Sagan (Żagań) palace with the Jesuit church, whose guardian was the duke, with a properly designed composition of greenery on a vast area obtained after the demolition of a part of the municipality buildings.

In 1549, the land of Sagan came directly under Habsburg rule. During the campaign of the Thirty Years' War, in 1627, Albrecht von Wallenstein appeared as one of the main commanders of the imperial army in Silesia. In 1628, the Emperor, who was constantly strengthening his court position, handed over the Duchy of Sagan as compensation for outstanding financial obligations [86] (pp. 59–68). Wallenstein started the reconstruction of the municipality destroyed during the warfare. In this almost completely Protestant center, he decided to create a center of Catholicism and a magnificent seat of power. The first manor house to be built was located at the New Market Square (present Słowiański Square). At the same time, Wallenstein started to build a palace in the place of a castle demolished for this purpose [86] (pp. 59–68). The residence, which combines defensive and representative functions, was built based on the designs of Vincenzo Boccacci, the duke's architect. The four-winged form with bastions in the corners surrounded by a moat referred to the Gothic castle, reproducing the Renaissance solutions of the Italian defense residences from the 16th century—*palazzo in fortezza*. The construction of the palace began in the spring of 1630. The ground floor and walls of the first floor of the building were erected on a three-meter-high platform. At the same time, the new duke expelled the Lutheran predecessors. He assigned an abandoned Franciscan monastery to the Jesuits. In 1632, the reconstruction of the municipality was disturbed by the war. Sagan was captured twice by General Arnim and the Swedish army. However, at the end of the year, Wallenstein regained Silesia. In 1633, the duke, who resided in Sagan, planned an extensive construction of a Jesuit college and a convent for a hundred alumni. The project was also developed by Vincenzo Boccaccio. The complex was to consist of two-story buildings added to the former Franciscan church, grouped around four inner courtyards. Only part of the work was carried out until Wallenstein suddenly died in 1634 [86] (pp. 82, 87–89).

In front of the palace in Sagan, which was built as a modern residence under Wallenstein, it was decided to create a forecourt by deliberately demolishing the buildings. The result was a large square which, together with the articulation of the palace's façade, was to create an urban system that was subordinated to typical Baroque rules. In order to provide a suitable perspective for its seat, Wallenstein ordered the demolition of over 70 houses, which was largely carried out [86] (pp. 63–68) [87,88]. Within this space, two composition axes were arranged, one facing the western façade, the other facing the northern, shorter and linked to the façade of the opposite tenement. The latter was overshadowed at the end of the 18th century by the palace of the duke's administration (Herzogliches Kammergebäude), a classicist building erected for the General Plenipotentiary of the Sagan Duchy (now the district court). The longer composition axis ran in the middle of an elongated square to the north of the municipality walls (currently occupied by the tower apartment buildings), created after the demolition. It was a representative entrance to the residence from the side of the square. The spacious square north of the mansion, obtained by removing four quarters of the buildings connecting with the New Market Square, was not finally arranged [5]. Perhaps there were plans to completely clean up the space between the palace and the Jesuit college founded by Wallenstein in the former Franciscan monastery stretching from the square along the municipality walls to the north, Figure 5. Undoubtedly, these were one of the broader plans to transform the municipality into a representative space in front of the residence. The open square had dimensions of almost 130 by 150 m, and the farthest perspective view of the palace opened from the intersection of the streets behind the New Market (currently at the intersection of Słowackiego and Teatralna Streets), i.e., at a distance of about 350 m. This allowed for the creation of exceptionally monumental

perspective compositions. Such extensive areas were probably planned as compositions with the use of greenery systems, tree espaliers and garden parterre, as it was done in the immediate vicinity of the palace at the turn of the 18th and 19th centuries. However, there is no design evidence or records of this. Indirectly, such plans may be evidenced by the care provided to the chateau garden surrounded by avenues with a zoo and a pheasantry garden [86] (p. 60).

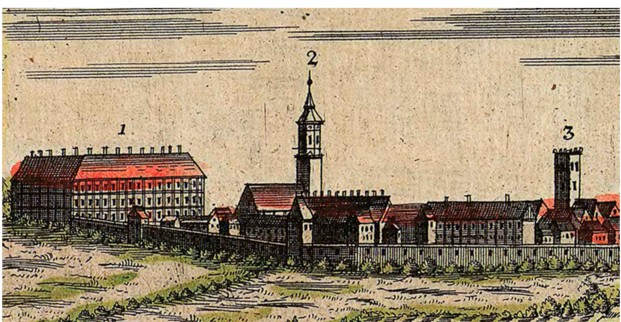

**Figure 5.** A fragment of the Sagan panorama showing a Jesuit palace and monastery with a church in the mid-18th century [87].

### 3.3. Tree Espaliers in Municipalities and Their Periphery

The green areas introduced into the municipality limits had the character of front garden parterres and tree espaliers, such as those in front of the residence in Bernstadt or Prusnitz, which were beginning to become an increasingly important compositional element. In Bernstadt, as it was probably mentioned in the 1680s, a line of linden trees protected the wall of the castle garden from the side of the municipality and marked the road serving the princely passage from the gate to the church. In a little town, Auras (Uraz), the line planted along the façade of a yuft factory founded in 1722 by Philipp Wilhelm Luther [89,90] and the accompanying residential buildings—the frontage of the market square and the street—marked the road to the bridge and the castle portal, Figure 6 [10] (p. 347). Wroclaw municipality-dweller Luther was granted an imperial patent for exclusive production of yuft in Silesia [91] and appropriate privileges for imported workers. He also took care of the hospital and suitable flats for them.

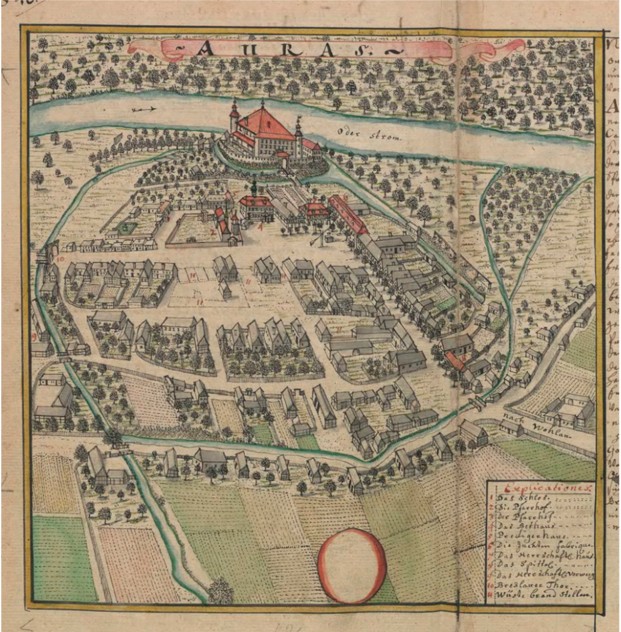

**Figure 6.** Auras in the middle of the 18th century in a drawing by F.B. Wernher [10] (p. 347).

Similarly to other European countries, tree espaliers and avenues were introduced in Silesia as early as in the 17th century. The lime avenue in Leubus (Lubiąż) was described as early as the end of the 17th century, so it had to be planted at least several dozen years earlier [23] (p. 1167). In Dyhernfurth (Brzeg Dolny), probably as early as 1666 or with the foundation of the Stations of the Cross after 1701, an allée connecting the residence with the sanctuary was planted [3] (pp. 47–50, 90–94) [4] (p. 75). Trees also lined roads—the old road to Wahren (Warzyń) and a new route from the village to the chapel and further towards the Odra River (led along the border of the medieval location department) [9] (pp. 410–411) [8] (p. 152). In Ohlau, a garden avenue was created in the 1670s, with walls in the form of a hedge or trellis, which was an extension of the composition axis of the residence behind the moat in the suburban gardens [77]. In Juliusburg, a palace and church garden and a road from the manor house were planted in rows [13] (pp. 700, 703). Regularly planted trees appeared on the fortifications of the Liegnitz (Legnica) castle [11] (pp. 7, 89). Avenues spread in Silesia, and lines of trees were created in the municipalities and around their walls [25]. Both smaller Silesian municipalities, such as Haynau, Ghurau, Kant, Nimptsch, Prochowitz, Steinau, Neumarkt, Goldberg and Landeshut (Chojnów, Góra, Kąty, Niemcza, Prochowice, Ścinawa, Środa, Złotoryja, and Kamienna Góra) [9] (pp. 559–560, 484–485) [11] (pp. 206–207, 208–209, 396–397) [14] (pp. 194–195) or much bigger such as Wolau, Ohlau, Bernstadt, Sagan, Oels [11] (pp. 396–397, 562, 567) [13] (pp. 688, 694–695) [87] or Schweidnitz (Świdnica) [15] (pp. 343–344) (A11 in Appendix A) [92]. They were surrounded by rings of fruit trees. Fortifications in Breslau (Wrocław) were provided with plantings on the southern side of the municipality [9] (pp. 196–199).

Even more widespread use was made of tree espaliers and avenues in the open landscape. They were commonly used in the complexes of residences, as shown in Wernher's drawings [9] (pp. 390–589) [11] (pp. 373–619) [12,13] (pp. 252–449, 662–681) [14]. They were used as a kind of shielding and decoration of monotonous walls or railroad fences and manor buildings of both extensive palace buildings and small manors. Such a line was planted in the aforementioned palace in Sybillenort. They can be seen in illustrations of numerous manor and palace complexes. Roads leading to palaces and manor houses were also transformed into avenues.

The tree espaliers were planted along the roads leading to the churches of Peace in Schweidnitz, Jauer (Jawor) (A12 in Appendix A) and Glogau [16] (p. 27) [93]. Three oak allées, of which fragments of one have survived to this day, leading to the gates in the wall surrounding the property of the Evangelical parish in Schweidnitz were built together with its construction, which is shown on plans from before the mid-18th century [94]. An avenue also led to the Church of Grace in Sagan [95]. All those suburban Evangelical churches of Peace and Grace were accompanied by cemeteries and schools and surrounded by lines of trees. Then, already in Prussian times, roads to the churches and their fences were provided with plantings. Similarly, trees were planted along the church walls, and the elevations of newly established Lutheran high schools, e.g., in Hirschberg (Jelenia Góra) [96], as shown in Wernher's drawings, which was in line with the state recommendations concerning the plantings surrounding schools and temples.

All main roads leading from Breslau began to be lined with trees already before the middle of the 18th century (A13 in Appendix A) [97–100]. At the turn of the 18th and 19th century, they transformed into highroads (*Kunststrassen*, *Chausseen*) 100 as well as many main roads in the whole region [101]. The Prussian national orders recommending the planting of trees decided that most of them took on the character of an avenue. However, the avenues did not become part of the composition of dense urban complexes of municipalities expanding over time. The avenue in Tschepin (Szczepin), shown on plans from the end of the 18th century, was not preserved in Breslau after it was incorporated into the urban layout [102], nor the famous Scheitniger (Szczytnicka) Avenue—a poplar avenue (virginischen Pappeln, Populus deltoides), planted in 1790, leading to the park founded by Friedrich Ludwig von Hohenlohe-Ingelfingen [103].

### 3.4. Avenues in Squares and Streets of Lower Silesian-Lusatian Border Municipalities

At first, in the neighborhood of Silesia, avenues started to play an important role in the layout and shaping of green urban interiors of municipalities. At that time, on the borderland of Lusatia and the Neumark, which lay beyond the borders of the Habsburg state, there were settlements related to the influx of religious fugitives, mainly from Silesia, valuable craftsmen, especially weavers. Many of the centers, which were established at that time as municipalities, lost their municipal rights quite quickly. Their beginnings, however, and thus their spatial organization, are connected with the process of founding the city.

In the medieval village of Trebschen (Trzebiechów), first mentioned in 1308 and situated in the Neumark, which in 1701 was incorporated into the Prussian state, a village that developed around the local Renaissance manor house belonging to the Troschke family from Züllichau (Sulechów) (A14 in Appendix A) [104] (p. 344) a Lutheran border church was built in 1654. Religious fugitives, among them weavers, began to come here. In 1680, a new large temple was erected [105]. The expanding village was granted municipality rights by a royal decree from 1707, and its then owner Konrad von Troschke named it after the first Prussian king Frederick I Hohenzollern (1657–1713)—Friedrichshuld (Frederick's Grace) [26,106].

The new buildings (in 1719, it was 25 houses) were erected along an avenue planted with double rows of lime trees, which connected the palace and the church (now Lipowa Street), Figure 7. The street was perhaps founded as early as 1680, and it certainly existed after 1707. It was designated as a composition axis connecting the entrance to the palace with the dominant of the church tower. The central part formed a kind of elongated square in the greenery, which was the center of the municipality. It was complemented by park avenues stretching north of the landowner's seat. The Baroque layout of the village is well readable today, although in 1823 a new building designed by K.F. Schinkel and his collaborators was erected on the site of the previous Evangelical temple, and the municipality houses were replaced after the fire of 1830, while the palace was rebuilt in the 1870s and 1880s, when Trebschen lost its municipal rights (1870) [9]. Linden alley was undoubtedly modelled largely on the Dresden Hauptstrasse as the closest example, probably well known to the Saxon courtier. However, the most important model was probably Berlin's Uner den Linden, as evidenced by the patron of the municipality, the first king in Prussia, mentioned in the name.

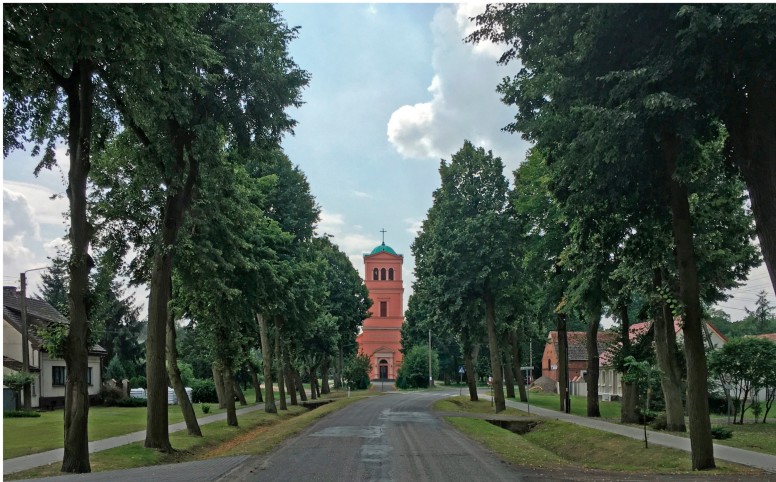

**Figure 7.** Aleja Lipowa (Linden Allée) in Trzebiechowo. Fot. Piotr Frąszczak, www.polska-org.pl (accessed on 27 October 2020).

In 1679, the old settlement of Dziadoszanie—Halbau (Iłowa) was granted municipality rights from the Elector of Saxony John George II (1613–1680) due to the request by Countess Helga Margaret von Friesen. The rights were valid until 1830, when the town was degraded

to a market municipality (Marktflecken) [27,107]. Its inhabitants were refugees from Silesia, who founded the first Evangelical parish in 1668 [108] (p. 836). The village developed in the hands of a new (since 1682) owner, Count Balthasar Erdmann von Promnitz, owner of Pless and Sorau (Pszczyna, Żary) [104] (pp. 101–102) [109], in the early 18th century. At that time, a rectangular market square was built up, with streets running out of corners, situated to the southeast of the land property. The erection of an evangelical parish church in 1720, designed by Giulio Simonetti [110,111], began after a fire in 1725, rebuilt and completed shortly before the next municipality fire, in 1749. At the same time, the Renaissance manor house from 1626 was converted and extended with a Baroque wing, giving it a palace-like form [108] (p. 836). Both buildings of the church and the palace, surmounted with towers protruding in front of the façades, were connected by a viewing axis, which in time transformed into a street surrounded by frontages [A15] [24,27]. At that time, around 1720, a second axial layout was also formed in Lindenstrasse (now K. Pułaskiego Street), located north of the first one, leading from the residence to the east (at its end in the 19th century a parish house was built up) [112–114] Figure 8 [20]. It was an extended avenue forming a kind of square planted with lime trees. In 1816, it was already an impressive avenue, so it must have been built several dozen years earlier [27]. The choice of trees indicates the Berlin pattern. However, it should be remembered that the founder knew the assumptions originating in Italy, France and the Netherlands [104] (p. 101). Greenery filling was not used in the case of the main urban interior—a marketplace, which was traditionally shaped like a square with streets leading from its corners; a second urban interior was added, probably related from the beginning to the area belonging to the Protestant community, where the school and parish buildings were erected [108] (p. 836).

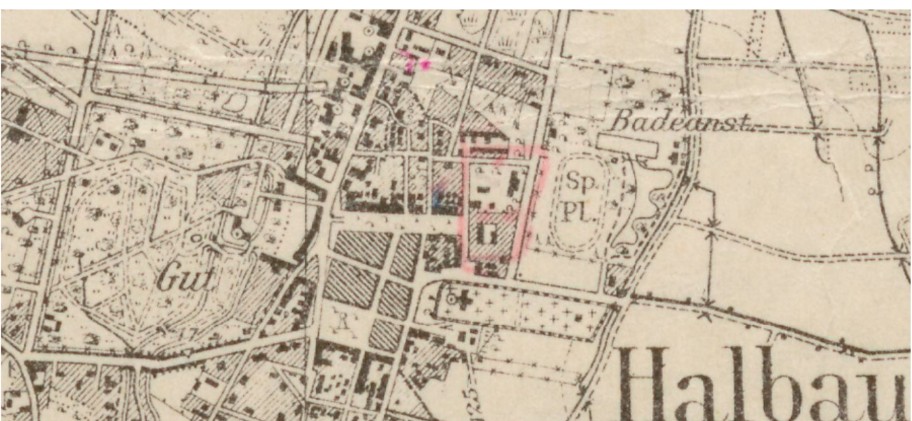

**Figure 8.** Halbau on a topographic plan from 1939 [113].

Within the borders of Silesia, which was taken over by the Prussian state, the first example of the introduction of urban greenery was its implementation of a new munici-pality established at that time—Neusalz (Nowa Sól). A significant economic problem of Lower Silesia was the lack of deposits of rock salt. On the initiative of Emperor Ferdinand I (1503–1564), attempts were made to become independent from the export of this product from Poland. As a result, the settlement of Neusalzburg (Neusalz) was established, where sea salt breweries began to operate from 1553, and which finally became a municipality in 1743 from the edict of King Frederick II of Prussia [2] (pp. 3–6, 29, 43 in, 136) [6] (pp. 8–17).

Neusalz, which initially developed as a settlement on the Oder River, was later given a Baroque irregular layout with the center in the vicinity of the salworks and the building of the salt office (later the town hall). Simultaneously with the granting of municipality rights, King Frederick II gave permission for religious freedom, thus allowing the settlement of the Moravian Brothers (Herrnhuter). On the route from Breslau to Grünberg (Zielona Góra), along the section of the road connecting Neustädtel (Nowe Miasteczko) and Wartenberg (Otyń), which was created with the formation of post riders, a colony was established,

equipped with a house of prayer, hospital, school, and pharmacy [2] (pp. 46, 49–52, 58) [6] (p. 19). Residential buildings, identical story houses, stood on both sides of an avenue planted with chestnut trees, Figure 9 [2] (p. 52) [14] (pp. 172–173). The choice of these trees, considered to be very ornamental, associated with superfluousness, is evidence of a desire to accentuate wealth and emphasize prestige. This shows a connection with Huguenot districts in German municipalities, but also with very representative projects, e.g., in Bayreuth and Ansbach.

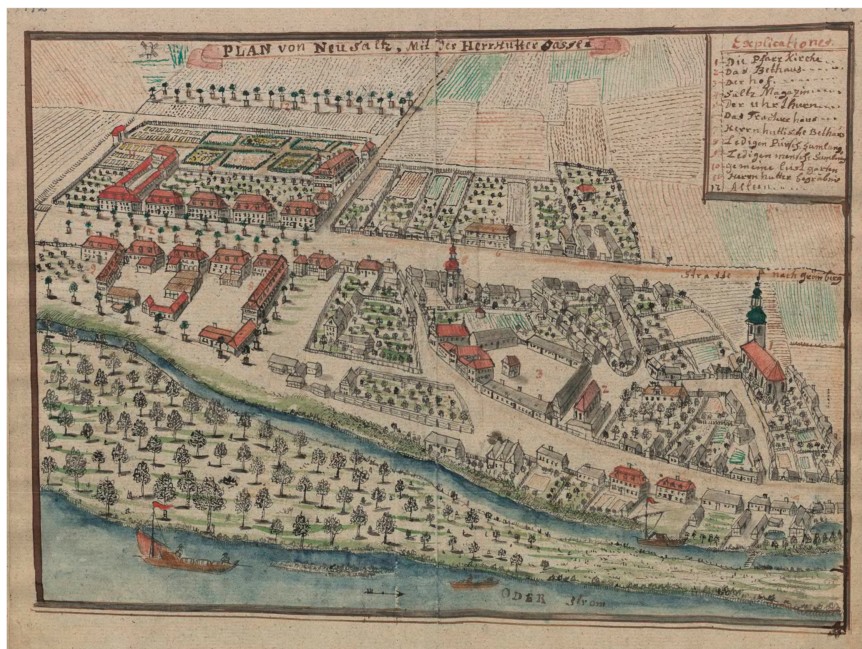

**Figure 9.** Neusaltz's perspective plan from the mid-18th century [14] (pp. 172–173).

After the 1759 fire during the Third Silesian War, it was rebuilt again according to strict rules from 1765 [2] (pp. 765, 73–75) [115]. The avenue existed in the 19th century, with the last trees being removed at the beginning of the 20th century [116], and its memory was immortalized by the name Kastanienallee (now Wrocławska Street).

The introduction of greenery into the center of the layouts has become a principle in the settlements of the Moravian Brothers. The Moravian (Czech) brothers under the care of Nikolaus von Zinzendorf received permission to settle in Upper Lusatia in Herrnhut in 1722 and then in Hesse in Herrnhaag in 1738 (A16 in Appendix A), in Niesky in 1742, and in 1756 in Kleinwelka near Bautzen. In Silesia, Frederick II allowed the construction of settlements in Neusaltz 1742, Gnadenberg (Godnów, currently in Kruszyn, near Bolesławiec), and Gnadenfrei near Reichenbach (Piława Górna, Dzierżoniów). Both settlements began to be built from 1743. Much later, in 1772, the Brothers also settled in Gnadenfeld (Pawłowiczki) in Upper Silesia [117–119]. Around the square with the house of prayer, their first settlement Herrnhut (1722) and another Herrnhaag (1738) were built, although they did not have municipality rights, but had a regular layout [7,8] (pp. 194–198).

Although the architecture of the main buildings of these settlements was designed by professionals such as architect Siegmund August von Gersdorff (1702–1777), a member of the community since 1743 [120] (pp. 109–112), their layout, according to recorded tradition, was the result of the ideas of the first spiritual leaders of the community. It is known that the Herrenhag settlement plan was drawn up by the then bishop of the community, Christian David Nitschmann (1696–1772) [121]. The concepts stemmed from the implementation of biblical principles considered to be the basis for the functioning of the community into not only European, but also American urban planning solutions known to the organizers.

In Herrnhut, the central square was occupied by a large building of the community house of prayer, the so-called Great Hall. In front of the house of prayer—opposite the

main axis of the complex marked outside the housing estate by an allée—a small front garden was made [122]. In Herrnhaag and Niesky, the houses of prayer were erected in frontages of squares, and their area was designated for extensive greenery, where rows of trees, connected by fences surrounded green parterres [123,124], Figure 10. In Herrnhaag, in the middle, a pavilion for a well was erected [125]. In Gnadenfrei Figure 11, Gnadenberg Figure 12, and Kleinwelka, initially the square was only decorated with trees, but as early as the beginning of the 19th century it was also surrounded by hedges [126]. This connects them not only with the history of the introduction of avenues in European municipalities, but also with the establishment of squares formed in a manner of a garden with the use of one-story greenery, the origins of which appeared in English municipalities and were a new element on the continent.

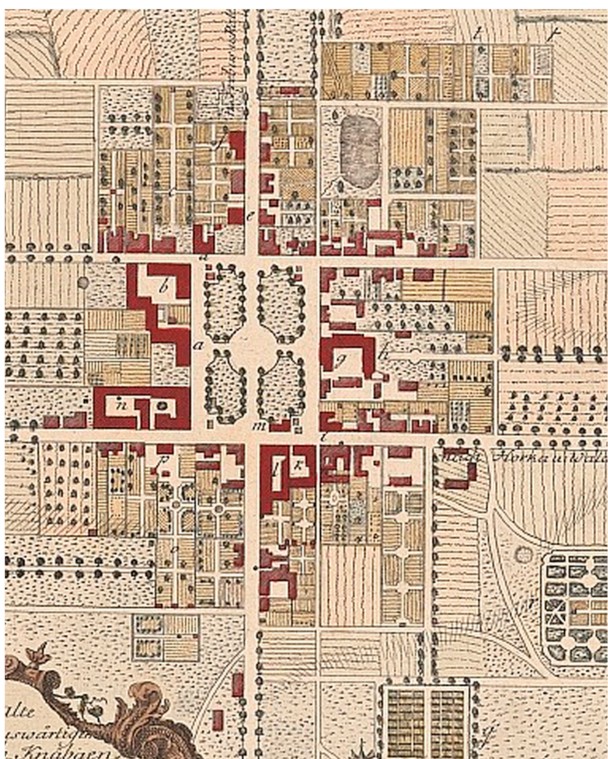

**Figure 10.** The Herrhaag plan from the time of its creation. Extract from the plan [122].

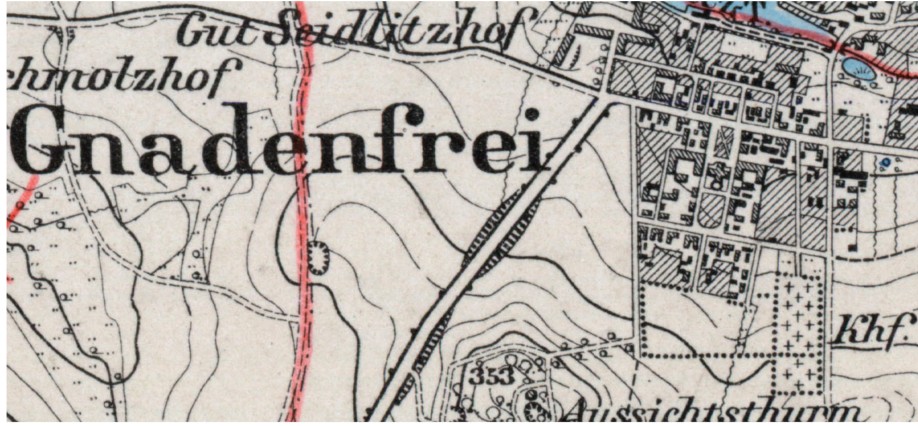

**Figure 11.** Gnadenfrei on a topographic plan from the early 20th century. Fragment of the plan [19].

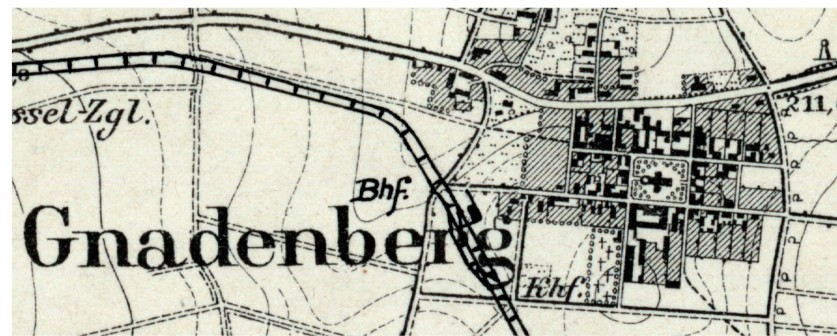

**Figure 12.** Gnadenberg of the topographical plan from the end of the 19th century. Fragment of the plan [17].

## 4. Discussion

In Lower Silesia, we associate urban greenery with the urban planning of the 19th century. However, it should be remembered that the origins of the use of gardens and trees in urban composition date back to modern times. Initially, in the municipalities of Lower Silesia and its borders, the only greenery was introduced by closed gardens. Decorative trees and shrubs were only visible from the windows of the palaces, the routes and loggias designed within them, and through the rails of the fences, as in front of the Brieg seat of George III, Christian and Louise of Anhalt. These were still largely viewing openings created in the Renaissance style. However, by this time, the role of greenery in forming the overall appearance of the residence was probably already beginning to be taken into account, which was an innovation in the design approach. In the process of shaping the foreground of the Baroque municipality residences, starting from the earliest examples created just after the Thirty Years' War in the middle of the 1670s in Ohlau and Wohlau, greenery compositions in the form of garden parterres became an element of entrance squares. The same was probably also planned for the unimplemented arrangements in Sagan. Slightly later, lines of trees were used to emphasize the façade and the procession road in front of the ducal seat in Bernstadt. In all these layouts, a long compositional axis was used—the most visible in the case of the Ohlau residence—leading to the palace portal through the middle of the garden plots, marked by the use of small architecture in the form of decorative bridges adorned with obelisks. At the back of the residences in Ohlau and Bernstadt, the axis was continued in the palace gardens, through an avenue with a trellis or a line of trees. In this way, a typical baroque solution of the palace layout was introduced, which was only beginning to be popular at that time. Moreover, the method of using greenery confirmed the knowledge of the most up-to-date patterns in the design of this type of complexes. This is clearly related to the founders, their education, and their connections. Louise of Anhalt and the dukes of both Württemberg lines belonged to the group of enlightened people and were widely related to European—German and French—aristocratic families. Scarce information concerning the designers of urban layouts—such as Vincenzo Boccaccio in Sagan, Carlo Rossi of Como in the case of the Ohlau residence, or Georg Herbst in the case of the gardens in Bernstadt [127]—indicates that only local artists were employed, which speaks all the more in favor of the importance of investors in the choice of design solutions. In these concepts, one can clearly trace the familiarity with French models, especially Versailles and the residences in Paris, which were being built at that time, and German models, those applied in the districts of Berlin or Dresden.

Alleys and tree rows had already become common in Lower Silesia and neighboring areas by the middle of the 18th century. In the urban layout, the lines surrounding the municipalities were incorporated into the fortification systems and they screened roads leading out of the cities. The avenues, which initially had a largely purely utilitarian significance, became the basic compositional skeleton of the new suburbs; the same happened in the intensively developing European cities of the time. In the open landscape, they were used to cover and accentuate roads, introducing optical partitions and Baroque viewing axes.

They were introduced into spatial arrangements creating perspective axes and shaping views and panoramas. In this way, the importance of the architecturally modest suburban Protestant churches of Peace and Grace and the accompanying schools was emphasized. The use of lines of trees was a kind of spiritual elevation. It also indicated ideological ties. The adoption of Dutch solutions through their use in German urban designs and districts for Huguenot refugees lent them an ideological sense in Lower Silesia. It signaled the Protestant affiliation of the emerging urban complexes. Tree espaliers and avenues were widespread and used for both utilitarian and aesthetic reasons; they also found application in urban interiors. In this case, too, they had not only an aesthetic significance but also a propaganda dimension. The tree espaliers in Auraz and the chestnut avenue in the district of the Moravian Brothers in Neusaltz highlighted the frontage of the buildings. At the same time, they indicated the area of the city inhabited by a separate Protestant community.

Avenues extended to elongated to green squares, they were the first to appear not in Silesia, where no new municipalities were created in the 18th century and no large urban districts developed, but in the border area of Lusatia and the Neumark at the beginning of the 18th century, in a period of cultural and political transition, initially still within the borders of the Wettin state, and later under Prussian rule, in close connection with the settlement of religious fugitives. In the case of these settlements, the existing arrangements based on traditional solutions with a rectangular square and streets running out of its corners were planned or adapted for expansion, but also Baroque axial systems connecting the residences with the square or the most important buildings of the villages were introduced. A specific novelty was the introduction of avenues planted with lime trees forming squares in Halbau and Friedrichshuld. The layout of new German urban districts for Huguenots and other religious refugees was modeled on the Berlin Dorotheenstadt and Friedrichsstadt with the Unter den Linden avenue. The form and prominence in the urban layout give them the expression of a green urban space, which is comparable to those shaped in the aforementioned cities in northern Brabant. In the municipalities founded by the Moravian brothers, green central squares were designed, in which not only trees were planted, but also lawns surrounded by fences were established. In this case, links can undoubtedly be made to the solutions used in cities and neighborhoods for religious exiles, starting with the first Renaissance layouts, through examples from various German countries. However, the way greenery was introduced into the interiors of the central squares also links them to the settlements of Protestant exiles in the neighboring areas mentioned in the article. The form of lines and green parterres, forming a kind of square—in principle not yet used on the continent (with the modest exception of the Vosges Square)—points to English inspiration. The fact that one of the communities was founded in London while Zinzendorf had contacts with and even made trips to that city would confirm the close links with that country. They thus become one of the earliest examples of the use of squares in European urban planning outside England.

## 5. Conclusions

These greenery compositions appeared in urban development in the region of Lower Silesia with some delay as compared to their popularization in western Europe, although the stages of their introduction are similar. The use of greenery in the layout of ducal urban residences began at the same time as this innovation spread across the continent, i.e., decades after the first similar developments. Even later, in relation to the earliest examples, tree rows were planted in Lower Silesia and border areas. The introduction of greenery into urban interiors was initially on a very limited scale. Green squares and inner municipality avenues appeared in the region later than in the Netherlands, France, Prussia, or other German countries. Patterns were undoubtedly drawn from these regions. It is worth remembering, however, that these were not only imitations of the activities of the Saxon and Prussian courts, but also, in part, were inspired by the solutions already applied in Lower Silesia. The ideas of the investors of the dukes, then the high courtiers, were accepted by religious communities. Thanks to this, they probably became a kind of

distinguishing feature of bourgeois culture, and utilitarian reasons decided to popularize avenues in the Prussian state.

Over time, the avenues became a distinctive feature of the landscape, preserved to this day in the areas of the former Neumark and Lusatia. In Lower Silesia, since the beginning of the 19th century, green promenades have been commonly created in areas after the demolition of fortifications (Breslau, Liegnitz, Glogau, Schwiednitz, Neisse but also, e.g., much smaller Bunzlau/Bolesławiec, Lüben/Lubin). The squares became popular in the 19th century and the beginning of the next one; in Breslau (A17 in Appendix A) and in all larger municipalities in newly planned squares in developing downtown districts (A18 in Appendix A). The tree espaliers were also planted within municipality centers along the market and street frontages. Such popularity and prestigious importance of the urban greenery had undoubtedly much to do with Baroque traditions.

Baroque urban solutions have left their mark on the urban layout to this day. Unfortunately, the arrangement of greenery spaces has largely disappeared. In residential municipalities, the compositions have undergone further transformations as in Oława or elimination as in Bierutów. Today, greenery arrangements are sometimes created, partly inspired by Baroque front gardens, for example in Brzeg. The Baroque avenues were removed from Uraz (along with the elimination of the whole municipality layout) and Nowa Sól after the expansion of the communication route. Of the municipalities founded by the Moravian Brothers, Herrnhaag existed for only a few decades and disappeared with the emigration of the community in 1753, the urban complex in Godnów was completely destroyed during the Second World War. This makes it all the more valuable for the present day to see the few arrangements that have survived to this day [128]. In Germany, the greenery complexes in Herrnhut and Niesky were revalorized and restored, while the lime trees were supplemented. In Trzebiechów, the composition of greenery with double lines (with trees evidenced in the 19th and 20th century) is largely preserved. Although in Iłowa the trees of the avenue have been mostly removed, the spatial arrangement has survived until today. The square in Piława Górna has turned into a modern green square without the rows of lime trees once surrounding it. It is worth understanding what the spatial impact was; these specific Baroque urban arrangements are worth protecting and revalorizing.

**Funding:** This research received no external funding

**Institutional Review Board Statement:** Not applicable.

**Informed Consent Statement:** Not applicable.

**Data Availability Statement:** Not applicable.

**Conflicts of Interest:** The authors declare no conflict of interest.

## Appendix A. Details and Data Supplemental to the Main Text

A1 Bocacio characterized: "per lo mezzo in assai parti vie ampissime; tutte diritte come strale e coperte di pergolatidi viti" [28]

A2 "Ambulacrum siue ambulatio, une allee", in a book for children from 1536 [31,32]

A3 "bellissima uista una strada diritta, ampia e polita... con gli arbori, iquali essendopiantati dall'una, e dall'altra parte con la uerdura allegrano glianimi nostri, e con l'ombra ne fanno commodo grandissimo" "Le vie fuori della Città si deuono far ampie, commode, con arbori d'amendue le parti; da i quali i uiandanti l'estate siano difesi dall'ardor del sole, e prendano gli occhi loro qualche ricreatione per la uerdura") [33] (pp. 262, 266)

A4 Some of these were removed as early as the eighteenth century.

A5 Edict of 1731, renewed in 1743 and rewritten in 1746, covered the protection of willow, mulberry and lime tree avenues and others. The edict recommended planting of forests with rows of lime trees [58,59].

A6 Already recorded in the 1603 urbarium.

A7 Then moved to the other side of the Oder River [8] (pp. 156–157).

A8 Perfectly representing the spatial relations due to the triangulation measurement points marked, the situation plan of the royal castle). From the municipality side, a square was formed along the entire palace façade—currently Castle Square [11] (pp. 267, 397, 400).

A9 It might be of interest to know how much trouble Wernher had with designing this entrance correctly. In each drawing, the portal and the bridge that precedes it is in a different place of the wing of Duchess Louise.

A10 The moat was probably eliminated in the 18th century.

A11 In 1769, it was already 560 trees [92].

A12 The alley is now reconstructed.

A13 Roads leading from Breslau to Ohlau via Tschechnitz (Siechnice), to Brieg, to Grottkau (Grodków), to Strehlen (Strzelin), to Schweidnitz via Klettendorf (Klecina) and Klein Tinz (Tyniec Mały), to Kant via Gräbschen (Grabiszyn), to Striegau (Strzegom) via Mochbern (Muchobór), to Neumarkt, Liegnitz and Glaogau via Lissa (Leśnica), to Auras and further to Trebnitz (Trzebnica) via Rosenthal (Różanka) and to Oels via Hundsfeld (Psie Pole).

A14 It was Konrad von Troschke (1638-1702)—between 1688-1691, leaseholder of the castle in Schwiebus (Świebodzin) and royal official—and his son Konrad (1671-1728), the last of the family) [104] (p. 344).

A15 In the second half of the 18th century, there were fewer than 500 inhabitants. Probably only the area around the marketplace, a place in the form of a square, later a new square, remained built-up at that time [24,27].

A16 The Herrnhuter temporarily expelled from Saxony settled in neighboring Hesse.

A17 The square: Am Wäldchen now Pomorska Street, Tauentzien Pl./T. Kościuszki Square, Matthiasplatz/Św. Maciej Sq., Kaiser Wilhelm Platz/Powstańców Roundabout.

A18 F.e in Liegnitz: Fridrichs Pl./pl. Słowiański, Bilze Pl./Skwer Orląt Lwowskich; in Waldenburg/Wałbrzych: Hermannsplatz/ Konstytucji 3 Maja Sq.; in Glogau: Wilhelm Pl. Currently the square of Rev. Z. Kutzan; in Hirschnerg; Wilhelms Pl./T. Kościuszki Sq., in Neisse: Vickoria Pl./Kopernika Sq.; in Oppeln/Opole: Friedrichs Pl./Daszyńskiego Sq.

*Source of Figures*

Biblioteka Uniwersytetu Wrocławskiego, Oddział Rękopisów. Wydział Geografii i Studiów Regionalnych Uniwersytetu Warszawskiego. Wydział Nauk Geograficznych Uniwersytetu Łódzkiego—all in Public Domain.

Sächsische Landesbibliothek-Staats- und Universitätsbibliothek Dresden SLUB/Deutsche Fotothek—Creative Commons Attribution-ShareAlike 4.0 International License.

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
