# Peer review of "Baroque Origins of the Greenery of Urban Interiors in Lower Silesia and the Border Areas of the Former Neumark and Lusatia"

_sustainability, doi:10.3390/su13052623_

Round 1
Reviewer 1 Report
The paper presents an interesting topic and relies on high originality. However, the research methodology is not properly emphasized, and the paper structure needs major improvements. The materials and methods section does not present any method - only an "analysis of local sources." I strongly suggest this analysis to be clarified so that a methodological framework is presented there. Otherwise, it could be considered as a "background" section. In this case, the paper would lack a methodology presentation. I would consider splitting results and discussion into two different sections. This work's contribution to the field would also be emphasized if its results were made clearer and articulated with its methods.
Reviewer 2 Report
Dear Editors,
I would like to thank You for opportunity of evaluation of the manuscript. The submitted manuscript titled „Baroque origins of the greenery of urban interiors in Lower Silesia and the border areas of the former Neumark and Lusatia” presents interesting results. However, I found some flawns, which should be improved before an eventual publication. In my opinion the chapter "Metodology" should be improved. Also the extensive chapter "Results and Discussion" is difficult to follow and understand.
Round 2
Reviewer 1 Report
The article was consistently improved in this review.
Reviewer 2 Report
Dear Author,
In my opinion text has been sufficiently improved and I do not have suggestions regarding further corrections.